# Targeting E2F Sensitizes Prostate Cancer Cells to Drug-Induced Replication Stress by Promoting Unscheduled CDK1 Activity

**DOI:** 10.3390/cancers14194952

**Published:** 2022-10-10

**Authors:** Mohaddase Hamidi, Ainhoa Eriz, Jone Mitxelena, Larraitz Fernandez-Ares, Igor Aurrekoetxea, Patricia Aspichueta, Ainhoa Iglesias-Ara, Ana M. Zubiaga

**Affiliations:** 1Department of Genetics, Physical Anthropology and Animal Physiology, University of the Basque Country UPV/EHU, 48080 Bilbao, Spain; 2Ikerbasque—Basque Foundation for Science, 48009 Bilbao, Spain; 3Department of Physiology, Faculty of Medicine and Nursing, University of Basque Country UPV/EHU, 48080 Bilbao, Spain; 4Biocruces Bizkaia Health Research Institute, 48903 Barakaldo, Spain; 5National Institute for the Study of Liver and Gastrointestinal Diseases (CIBERehd, Instituto de Salud Carlos III), 28029 Madrid, Spain

**Keywords:** E2F, replication stress, nucleotide biosynthesis, CDK, prostate cancer, apoptosis, ATR

## Abstract

**Simple Summary:**

E2F1 and E2F2 are highly expressed in many cancer types, but their contribution to malignancy is not well understood. Here we aimed to define the impact of E2F1/E2F2 deregulation in prostate cancer. We show that inhibition of E2F sensitizes prostate cancer cells to drug-induced replication stress and cell death. We found that E2F target genes involved in nucleotide biosynthesis contribute to maintaining genome stability in prostate cancer cells, but their enzymatic activity is insufficient to prevent replication stress after E2F1/E2F2 depletion. Instead, E2F1/E2F2 hinder premature CDK1 activation during S phase, which is key to ensure genome stability and viability of prostate cancer cells. From a therapeutic perspective, inhibiting E2F activity provokes catastrophic levels of replication stress and blunts xenograft growth in combination with drugs targeting nucleotide biosynthesis or DNA repair. Our results highlight the suitability of targeting E2F for the treatment of prostate cancer.

**Abstract:**

E2F1/E2F2 expression correlates with malignancy in prostate cancer (PCa), but its functional significance remains unresolved. To define the mechanisms governed by E2F in PCa, we analyzed the contribution of E2F target genes to the control of genome integrity, and the impact of modulating E2F activity on PCa progression. We show that silencing or inhibiting E2F1/E2F2 induces DNA damage during S phase and potentiates 5-FU-induced replication stress and cellular toxicity. Inhibition of E2F downregulates the expression of E2F targets involved in nucleotide biosynthesis (*TK1*, *DCK*, *TYMS*), whose expression is upregulated by 5-FU. However, their enzymatic products failed to rescue DNA damage of E2F1/E2F2 knockdown cells, suggesting additional mechanisms for E2F function. Interestingly, targeting E2F1/E2F2 in PCa cells reduced *WEE1* expression and resulted in premature CDK1 activation during S phase. Inhibition of CDK1/CDK2 prevented DNA damage induced by E2F loss, suggesting that E2F1/E2F2 safeguard genome integrity by restraining CDK1/CDK2 activity. Importantly, combined inhibition of E2F and ATR boosted replication stress and dramatically reduced tumorigenic capacity of PCa cells in xenografts. Collectively, inhibition of E2F in combination with drugs targeting nucleotide biosynthesis or DNA repair is a promising strategy to provoke catastrophic levels of replication stress that could be applied to PCa treatment.

## 1. Introduction

E2F activity controls the transcriptional regulation of a myriad of genes encoding key proteins involved in nucleotide biosynthesis, DNA replication and DNA repair [1,2]. E2F target gene expression is regulated by the temporal association of E2F proteins with members of the Retinoblastoma family of tumor suppressors (RB) [3]. RB binds to E2F and keeps E2F-dependent transcription inactive. Activation of E2F-dependent transcription is initiated by growth signals that stimulate CDK4/6 and CDK2 activities. These kinases promote sequential inhibitory phosphorylations on RB proteins, leading to their dissociation from E2F factors. Free E2F can now induce the transcription of its target genes. Within the mammalian E2F family (E2F1-8), E2F1 and E2F2 are included in the “activator” subset, owing to their ability to activate transcription and cellular proliferation in culture [4]. In vivo, E2F1 and E2F2 are key mediators of genomic stability maintenance, as chronic depletion of E2F1/E2F2 in mice results in high levels of replication stress, DNA damage and subsequent apoptosis and senescence [5,6]. The mechanisms underlying this phenotype remain to be elucidated, but it highlights the importance of keeping E2F activity under control to ensure cell viability.

E2F1 and E2F2 are upregulated in several cancer types, including prostate cancer (PCa) [7,8], in which overexpression of these E2Fs correlates with poor clinical outcome [9,10]. In this regard, it has been suggested that cancer cells rely on E2F function to tolerate the exceptionally high levels of basal or drug-induced DNA replication stress that cancer cells endure [11,12]. Replication stress induces the expression of E2F target genes involved in checkpoint control, DNA repair and nucleotide biosynthesis, which have been proposed as mediators of the recovery from this type of stress and subsequent DNA damage [13]. Thus, this dependency of cancer cells on E2F activity might provide new therapeutic opportunities in combination with the current standard treatments.

In response to DNA replication stress, cells activate an intra-S checkpoint that promotes replication fork slowing/stalling and cell cycle delay. The kinase ATR and its effector kinase CHK1 are known to orchestrate this checkpoint, which controls replication origin firing, stabilizes damaged replication forks and delays cell cycle progression in order to create a time window to resolve DNA lesions [14,15]. Central to this response is the inactivation of CDK1, a kinase whose activity is critical for driving cells into mitosis [14,15]. Downregulation of CDK1 activity is achieved through its inhibitory phosphorylation at Tyr15, carried out by the concerted activity of kinases and phosphatases, including WEE1 and CDC25A, both E2F transcriptional targets [16]. An aberrant CDK1 activity during S phase has been reported to cause unscheduled firing of DNA synthesis, replication stress and widespread DNA damage, which further activates the ATR pathway [17,18,19]. Not surprisingly, recent clinical trials targeting key mediators of this pathway (WEEI, ATR, CHK1), in combination with classical chemotherapy aiming to induce catastrophic DNA damage, are starting to show promising results in cancer treatment, including PCa [20] (https://clinicaltrials.gov/ct2/show/NCT03787680).

Standard chemotherapy based on nucleoside analogs is widely used in the treatment of several malignancies. Their cytotoxicity is typically caused by enhanced replication stress following an imbalance of the cellular nucleotide pool [21,22], but tumor cells eventually become resistant to this type of treatment. PCa is particularly resistant to the treatment with nucleoside analogs, such as 5-fluorouracil (5-FU) [23], an antimetabolite that inactivates the enzymatic activity of Thymidylate Synthase (TS), the key enzyme for de novo synthesis of 2′-deoxythymidine-5′-monophosphate (dTMP) [24]. Although the mechanisms of resistance to nucleoside analog drugs have not been sufficiently clarified, overexpression of E2F target genes involved in the metabolism of nucleotides may play a role in this resistance [22,25].

An understanding of the mechanisms that underlie the impact of inhibiting E2F activity in cancer treatment remains elusive. Here we show that targeting E2F1/E2F2 elicits DNA damage during S phase, leading to premature CDK1 activation and compromised PCa viability. We further show that genetic or chemical inhibition of E2F1/E2F2 potentiates replication stress and cellular toxicity induced by 5-FU or by ATR inhibition in PCa cells through mechanisms involving transcriptional deregulation of key E2F responsive genes involved in the recovery from genotoxic damage. Thus, targeting E2F may be a valuable strategy in PCa treatment, in combination with drugs that restrict nucleotide biosynthesis or DNA repair.

## 2. Materials and Methods

### 2.1. Cell Lines, Drugs and Reagents

PC3 and DU145 PCa cell lines were a kind gift from Drs. Arkaitz Carracedo and Veronica Torrano. Cells were maintained in Dulbecco’s modified Eagle’s medium supplemented with fetal bovine serum (10%). The compounds used for this study were: 5-fluorouracil (5-FU; Sigma-Aldrich, St. Louis, MO, USA), E2F inhibitor HLM006474 (E2Fi; Merck Millipore, Darmstadt, Germany), CDK1 inhibitor RO-3306 (CDK1i; Merck Millipore), CDK2 inhibitor CAS 222035-13-4 (CDK2i; sc-221409, Santa Cruz, CA, USA), ATR inhibitor AZD6738 (ATRi; SelleckChem, Houston, TX, USA), dNMPs (dAMP, dTMP, dGMP, dCMP Sigma Aldrich), 5-bromo-2′-deoxyuridine (BrdU; Sigma-Aldrich), Propidium Iodide (PI; Sigma-Aldrich), RNAse A (Sigma-Aldrich).

### 2.2. Transfections and siRNA-Mediated Knockdown

For knockdown of endogenous *E2F1*, *E2F2*, *TK1* and *DCK*, two different sequences of commercial small interfering RNA (siRNA) oligonucleotides (Life Technologies, Carlsbad, CA, USA) were transfected at a final concentration of 7.5 nM using Lipofectamine RNAiMAX (Life Technologies) following the manufacturer’s recommendation (Appendix A).

### 2.3. Analysis of Cell Cycle Distribution

To assess cell-cycle distribution, cells were fixed with chilled 70% ethanol in PBS, stained with 50 µg/mL propidium iodide (PI) and 5 µg RNAse A and analyzed by Attune-NxT (Thermo Fisher Scientific, Waltham, MA, USA) flow cytometer. For flow cytometry analysis of S-phase, cells were kept in culture with thymidine analog BrdU 10 μM for the last 2 h of culture. At the indicated time-points, cells were trypsinized and fixed with chilled 70% ethanol. Fixed cells were treated with 2M HCl and 0.1% Triton-X100 for 30 min for DNA denaturation and neutralized by adding Na_2_B_4_O_7_ 0.1 M pH 8.5. Cells were incubated with anti-BrdU antibody (M0744, Dako, Glostrup, Denmark) diluted 1:100 in 0.1% Tween-20/2% BSA solution for 1 h. R.T. Samples were washed with permeabilization solution (0.1% Tween-20), followed by incubation with the secondary antibody against mouse immunoglobulin labeled with a green fluorophore (Alexa Fluor 488) (A11001, Invitrogen, Life Technologies, Carlsbad, CA, USA) for 1 h at room temperature. Subsequently, cells were stained with PI as described above. Samples were analyzed using Attune NxT flow cytometer and data were processed with the Attune NxT Software (Thermo Fisher Scientific, Waltham, MA, USA).

### 2.4. Cytometric Analysis of γH2AX, Phospho-MPM2 and Phoshpo-H3

Cell cultures were ethanol-fixed, centrifuged and permeabilized with 0.1% Tween-20 in PBS. Subsequently, cells were incubated for 2 h at room temperature with a specific antibody against γH2AX (05-636, Millipore) diluted 1:500 in 0.1% Tween-20/2% BSA solution. Samples were washed twice with permeabilization solution (0.1% Tween-20), followed by incubation with the secondary antibody against mouse immunoglobulin labeled with a green fluorophore (Alexa Fluor 488) (A11001, Invitrogen), for 1 h at room temperature in the dark. After incubation with the secondary antibody, two washes were performed with permeabilization solution to finally stain the DNA with PI. Samples were analyzed with an Attune NxT flow cytometer (Thermo Fisher). Single cell data from cytometry files were obtained using a custom Python script.

For functional measurement of CDK1 activity, cells were fixed with chilled 70% ethanol and permeabilized with 0.1% Tween-20 in PBS. Incubation with anti-phospho-MPM2 antibody (05-638, Merck Millipore) diluted 1:1000 or phospho-H3 (06-570, Merck) diluted 1:500 in 0.1% Tween-20/2% BSA solution was carried out for 2 h R.T. Cells were washed twice with 0.1% Tween-20 in PBS and incubated with the corresponding secondary antibody labeled with a green fluorophore (Alexa Fluor 488) for 1 h R.T. protected from light. Two washes were performed after the incubation, followed by staining with PI and RNAse A. Samples were analyzed with an Attune NxT flow cytometer (Thermo Fisher). Single cell data from cytometry files were obtained using a custom Python script.

### 2.5. Crystal Violet Colony Formation Assay (CFA)

Cells were treated as indicated in each case, kept in culture for 24 or 72 h, and drugs were removed by adding fresh media. After 10–14 days in culture, surviving cell colonies were fixed with 4% of paraformaldehyde in PBS for 10 min, stained with 0.1% crystal violet (in 70% ethanol/PBS) for 15 min, and washed twice with PBS. Finally, pictures of each well were taken, and colonies were solubilized using 20% acetic acid. Samples were collected and absorbance was measured in triplicate for each condition at 590 nm using a spectrophotometer (Magellan, Tecan, Männendorf, Zürich, Switzerland).

### 2.6. Mouse Xenograft Experiments

Protocols for animal experiments were approved by the University of the Basque Country Animal Care and Use Committee (M20_2021_128, date of approval 13 of September 2021) and were strictly followed. Male athymic nude mice (Crl:CD1-Foxn1nu, ref. 086), 5 weeks old, were purchased from Charles River Laboratories. PC3 cells (3 × 10^6^ cells) in passage 8 were injected subcutaneously into both flanks in a volume of 100 μL (equal proportions of DMEM and Matrigel (BD Bioscience, Franklin Lakes, NJ, USA). Cells were tested for mycoplasma (Lonza MycoAlert Mycoplasma Detection Kit, Basel, Switzerland) prior to inoculation in mice. Tumor diameter was obtained by using a caliper (Vernier Caliper, VWR, Radnor, PA, USA) and tumor volume was calculated as 4/3 π r^3^. Mice began receiving treatment once tumors reached approximately 65 mm^3^ (±15%). Drugs were administered intraperitoneally (IP) at 25 mg/kg (5-FU), 50 mg/kg (E2Fi) and 25 mg/kg (ATRi) every 2 to 3 days. The dosing volume was 2 mL/kg. The experimental endpoint was defined as the day in which any single tumor reached 2000 mm^3^.

### 2.7. Quantitative RT-PCR Analysis

Total RNA was isolated using NZY Total RNA Isolation kit (Nzytech, Lisbon, Portugal). cDNA was synthesized from 1 μg of RNA using High-Capacity cDNA Reverse Transcription Kit (Thermo Fisher, Foster City, CA, USA), and all samples were diluted to the same final cDNA concentration. Real-time PCR was performed on several cDNA dilutions plus 1× SYBR green PCR Master Mix (Thermo Fisher Scientific) and 300 to 900 nM of primers for the analyzed genes (sequences in Appendix A). Reactions were carried out in triplicate using a QuantStudio 3 (Thermo Fisher Scientific) thermocycler for 40 cycles (95 °C for 15 s and 60 °C for 1 min) after an initial 10 min incubation at 95 °C. Relative amounts of cDNA were normalized to the internal control L-19, whose levels were invariant in all the analyzed conditions.

### 2.8. Protein Extraction and Western Blot Analysis

Cellular extracts were prepared as previously described [5]. Protein concentrations in supernatants were determined using CD Protein Assay (Bio-Rad Laboratories, Hercules, CA, USA). We performed Ponceau S staining to confirm correct protein transfer. Western blots were performed with 20–30 μg of total protein extract, using antibodies against: E2F1 (1:400, sc-256 Santa Cruz, CA, USA), E2F2 (1:400, sc-633 Santa Cruz), TK1 (1:1000, A5-29686 Invitrogen), DCK (1:400, sc-393099 Santa Cruz), TS (1:400, sc-3930945 Santa Cruz), P-CHK1 Ser345 (1:1000, 2348 Cell Signaling, Danvers, MA, USA), CHK1 (1:400, sc-7898 Santa Cruz), P-RPA Ser4/8 (1:1000, A300-245A Bethyl, Waltham, MA, USA), RPA (1:1000, ab2175 Abcam, Cambridge, UK), Phospho-CDK1 Tyr15 (1:1000, 9111 Cell Signaling), CDK1 (1:1000, ab18 Abcam), HSP90 (1:2000, sc-13119 Santa Cruz). Immunocomplexes were visualized with horseradish peroxidase-conjugated anti-mouse (1:4000, sc-3697 Santa Cruz) or anti-rabbit (1:4000, sc-2030 Santa Cruz) IgG antibodies, followed by chemiluminescence detection (ECL, Amersham, GE Healthcare Bio-Sciences, Pittsburgh, PA, USA) with a ChemiDoc camera (Bio-Rad Laboratories, Hercules, CA, USA). Densitometry-based quantification was performed using Fiji software. Relative optical density was calculated by dividing the densitometry of the protein of interest with the respective loading control.

### 2.9. Bioinformatics Analysis

CANCERTOOL (http://web.bioinformatics.cicbiogune.es/CANCERTOOL/index.html; accessed on: 29 April 2022) is an open-access resource for the analysis of gene expression and functional enrichments in different types of cancer, including prostate cancer [26]. Here, it was used to compare expression levels of *E2Fs*, *TK1*, *DCK1* and *TYMS* in PCa. A Student’s *t*-test was performed in order to compare the mean gene expression between two groups. We also analyzed the relationship between *E2Fs*, *TK1*, *DCK1* and *TYMS* expression levels in PCa and disease-free survival (DFS). A Mantel–Cox test was performed in order to compare the differences between curves, while a Cox proportional hazards regression model was performed to calculate the Hazard Ratio (HR) between the groups analyzed.

### 2.10. Statistical Analysis

GraphPad Prism 8.0 (GraphPad Software, San Diego, CA, USA) was used for statistical analysis and data representation. Data are given as mean ± SD. Statistical analysis was performed using ANOVA and Fisher’s test. Significance was defined by *p* < 0.05.

## 3. Results

### 3.1. Targeting E2F1/2 in PCa Cells Induces Replication Stress and Compromises Cellular Survival

Prostate tumor specimens derived from patients display upregulated levels of *E2F1* and *E2F2*, which are positively correlated with tumor malignancy, especially in the samples with the highest score in Gleason grade, and negatively correlated with disease-free survival (Appendix A). Hence, PCa may be particularly sensitive to E2F inhibition. Consistent with this notion, depletion of E2F1 and E2F2 by RNA interference in the PC3 prostate cancer cell line significantly raised the percentage of Serine 139-phosphorylated histone H2AX (γH2AX)-positive cells, indicative of an increased DNA damage signaling (Figure 1A, Appendix A). Furthermore, treatment with the E2F inhibitor HLM006474 (E2Fi), which blocks E2F transcriptional activity [27], induced similarly high levels of DNA damage in two prostate cancer cell lines, PC3 and DU145 (Figure 1B). Of note, E2Fi reduced dramatically the expression of E2F1 and E2F2 as well as other cell cycle genes (Appendix A), all of which are transcriptional targets of E2F, consistent with the reported role of this molecule as inhibitor of E2F activity [27]. DNA damage checkpoint activation after genetic (siRNA) of chemical (E2Fi) inhibition of E2F1 and E2F2 was further evidenced by the relative increase of phospho-RPA and phospho-CHK1 levels compared with the unphosphorylated proteins (Figure 1C,D).

We analyzed the cell cycle distribution and replication capacity of PC3 cells pulsed with BrdU upon E2F1/E2F2 knockdown. As expected, depletion of E2F1/E2F2 reduced dramatically the fraction of cells actively incorporating BrdU (Figure 1E, green rectangles), consistent with their role as promoters of DNA replication. Remarkably, we detected an accumulation of BrdU-negative cells with a DNA content that was intermediate between 2C and 4C in the siRNA-treated cells compared to siCtrl (Figure 1E, red circles). Treatment with E2Fi produced a similar reduction in BrdU-positive cells and an accumulation of nonreplicating cells in S-phase (Figure 1F). This evidence suggests that DNA damage after E2F1/2 inhibition activates an intra-S phase checkpoint. Importantly, E2F1/2 knockdown or exposure to E2Fi in PC3 cells was accompanied by a decrease in cellular viability (Figure 1G,H). Similar results were found with DU145 cells (Figure 1I). Altogether, these results indicate that E2F1 and E2F2 play a critical role in preserving genome integrity during S-phase in prostate cancer cells.

### 3.2. Depletion of E2F1/2 Potentiates 5-FU-elicited Cytotoxicity through Modulation of Target Genes Involved in Nucleotide Biosynthesis

Nucleotide analogs, widely employed in chemotherapy, function by targeting enzymatic activities involved in nucleotide biosynthesis [21], many of which are known to be transcriptionally regulated by E2F. We asked whether targeting E2F1 and E2F2 would affect cytotoxicity induced by these antimetabolites in prostate cancer cells. Interestingly, we found that silencing of E2F1/E2F2 in PC3 and DU145 cells potentiates 5-FU-induced accumulation of γH2AX (Figure 2A, Appendix A), and reduced the percentage of viable cells in culture by around 50–60% (Figure 2B,C). Similar results were found with a second set of siRNA oligos (Appendix A). Consistent with these findings, transfection with siE2F1/siE2F2 molecules in combination with 5-FU addition increased the fraction of apoptotic cells (measured as the subG1 subset by flow cytometry) to 50% from the nearly 30% apoptotic rate observed with the individual treatments (Figure 2D). Importantly, combination of 5-FU and E2Fi drastically reduced colony forming capacity of PC3 and DU145 cells (Figure 2E,F, Appendix A) and efficiently reduced xenograft growth of PC3 cells in mice without affecting animal wellbeing (Appendix A), corroborating the strong antiproliferative effect of the combinatorial treatment. These results support the notion that targeting E2Fs sensitizes prostate cancer cells to 5-FU-induced cytotoxicity.

We next investigated the mechanisms underlying the impact of targeting E2F in combination with 5-FU. We evaluated the expression of E2F1/2 and their transcriptional targets involved in nucleotide biosynthesis after addition of 5-FU to the cells. Exposure to 5-FU led to a transient increase in *E2F1* and *E2F2* mRNA levels with a peak at the 24 h time-point. Interestingly, this was accompanied by the transient expression of *GMPS* and *RRM2* at 24 h, and the sustained expression of *TYMS*, *TK1* and *DCK* both at 24 and 72 h (Figure 3A,B).

We assessed the role of E2F in the sustained upregulation of nucleotide synthesis genes upon 5-FU exposure by transfecting PC3 and DU145 prostate cancer cells with two different sets of siRNA oligos specific for E2F1 and E2F2, and 24 h later treated the cells with 5-FU or vehicle for 72 h. Silencing of E2F1, and, more potently, combined silencing of E2F1 and E2F2, prevented 5-FU-induced increase in TK1, DCK and *TYMS* (TS) mRNA and protein levels (Figure 3C,D and Appendix A). Moreover, treatment of prostate cancer cells with E2Fi recapitulated the results obtained after siRNA knockdown of E2F1/E2F2 in combination with 5-FU treatment (Figure 4A–C).

The product encoded by *TYMS*, Thymidylate Synthase (TS), is a component of the de novo pathway of nucleotide synthesis, and its activity is specifically inhibited by 5-FU. By contrast, TK1 and DCK are components of the salvage pathway of nucleotide biosynthesis, considered an alternative branch for nucleotide production that can overcome nucleotide pool reductions when the de novo pathway is inactivated [25]. With this in mind, we next assessed whether downregulation of TK1 and DCK levels could account for the increased replication stress and the potentiation of 5-FU-elicited effects observed after depletion of E2F1/2 (Figure 5). 

Combined knockdown of TK1/DCK in PCa cells induced a modest replication stress response (Figure 5A,B, Appendix A), and addition of the deoxynucleotide monophosphate (dNMP) products of TK1, DCK and TS activities dose-dependently recovered basal γH2AX levels in 5-FU-treated cells (Figure 5C). However, they failed to counteract DNA damage in E2F1/E2F2 knockdown cells, regardless of the presence of 5-FU (Figure 5C). These results indicated that E2F1 and E2F2 safeguard genome integrity through mechanisms that are independent of its target genes *TK1/DCK/TYMS*.

### 3.3. E2F1 and E2F2 Restrain CDK1 Activity during S Phase to Prevent DNA Damage

In non-stressed cells, the activity of the cell cycle regulator CDK1 is tightly regulated during S-phase, through its inhibitory phosphorylation at Tyr15, in order to ensure that DNA replication is completed before mitosis. Timely loss of this phosphorylation provides the cells the signal to enter into mitosis [28]. However, premature loss of CDK1 phosphorylation in S-phase frequently results in excessive replication stress [29]. Given the aberrant levels of replication stress in E2F-depleted cells, we examined the extent of CDK1 phosphorylation under these conditions. Strikingly, we found that E2F1/2 knockdown or E2F inhibition significantly reduced the levels of phospho-CDK1 relative to total levels of CDK1 in PC3 and DU145 cells (Figure 6A,B, Appendix A). By contrast, this reduction was not observed with combined TK1/DCK silencing, regardless of the presence of 5-FU in the cultures (Appendix A), suggesting that E2F1 and E2F2 modulate phosphorylation of CDK1 through TK1/DCK/TS-independent mechanisms.

The kinase encoded by the E2F target gene *WEE1* promotes the inhibitory phosphorylation of both CDK1 and CDK2 [19,30]. Importantly, *WEE1* mRNA levels were reduced in E2F1/2 knockdown or E2Fi-treated cells (Figure 6C,D). The sustained loss of phospho-CDK1 inhibitory phosphorylation (Tyr15) prompted us to assess CDK1 activity upon E2F inhibition, using pMPM2 as a surrogate marker of CDK1 and CDK2 activities [31,32]. We observed a clear reduction in the percentage of mitotic pMPM2-positive cells in E2Fi-treated samples (Appendix A), indicating that E2F-depleted cells are not entering mitosis, and consistent with the intra-S phase checkpoint observed upon E2F loss. By contrast, E2F1/2-depleted or E2Fi-treated cells exhibited increased intensity in pMPM2 signal when the subset of cells in S-phase was analyzed (Figure 6E,F, Appendix A). Increased CDK1 activity upon E2F loss was additionally supported by the increased intensity in pH3 signal in S-phase E2Fi-treated cells (Appendix A).

These findings raised the possibility that the DNA damage detected after E2F1/E2F2 inhibition could be attributed to premature CDK1 or CDK2 activation during S phase. Indeed, chemical inhibition of CDK1 with RO3306 [33] reduced the accumulation of γH2AX that is observed upon E2Fi treatment in PC3 cells (Figure 6G). CDK2 inhibition did not affect the levels of E2Fi-induced DNA damage in these cells (Appendix A). By contrast, E2Fi-elicited DNA damage was not recovered following CDK1 inhibition in DU145 cells, whereas CDK2 inhibition reduced significantly γH2AX levels in these cells (Appendix A), suggesting different dependencies on CDK1 and CDK2 activities in PC3 and DU145 cell lines. Altogether, these data indicate that E2F1 and E2F2 preserve genome integrity through the regulation of an intrinsic replication stress checkpoint during S-phase necessary for correct cell cycle progression and genome integrity preservation.

### 3.4. Combining E2F and ATR Inhibitors Boosts Replication Stress and Prevents Prostate Cancer Cell Growth in Mice

In a situation of replication stress, cells initiate a cascade of metabolic pathways orchestrated by ATR kinase in order to guide the repair of damaged DNA [34]. Small molecule ATR kinase inhibitors developed recently efficiently hinder DNA repair and thus potentiate the effect of DNA-damaging drugs [35]. Our findings that genetic or chemical inhibition of E2F1/E2F2 induce the accumulation of several DNA damage markers (γH2AX, pRPA, pCHK1) evidence an activation of the ATR pathway in E2F-targeted cells. We asked whether inhibiting ATR with the specific small molecule inhibitor AZD6738 (ATRi) could potentiate E2Fi-elicited cytotoxic effects in prostate cancer cells. Treatment with ATRi alone had a negligible effect on γH2AX accumulation in prostate cancer cells, and E2Fi alone induced detectable γH2AX levels. Importantly, the combination of E2Fi and ATRi led to a significant increase in γH2AX intensity in PC3 and DU145 cells compared to the single treatments (Figure 7A, Appendix A). Moreover, the increased in γH2AX upon E2Fi and ATRi combination in PC3 was alleviated when cells were treated with CDK1i (Figure 7B), supporting that the additive effect between E2Fi and ATRi depends on CDK1. Additionally, the combination of E2Fi and ATRi drastically reduced colony forming capacity of PC3 cells (Figure 7C). Prompted by the finding that loss of E2F activity sensitizes cells to ATR inhibition, we examined whether the cytotoxic effect exhibited by the combination of E2Fi and ATRi could be recapitulated in vivo, by assessing PC3 tumor growth in nude mice treated with these drugs. Administration of either drug individually did not lead to a significant reduction in tumor volume. Conversely, PC3 xenograft growth in nude mice was blunted substantially by the combination of E2Fi and ATRi, without affecting animal wellbeing (Figure 7D–F, Appendix A). Taken together, our results provide support for inhibiting E2F activity in combination with ATR inhibition as a promising strategy to provoke catastrophic levels of replication stress in tumor cells that deserves further investigation.

## 4. Discussion

High levels of replication stress are a common feature in cancer cells, and it is assumed that the onset of replication stress represents an early event during tumorigenesis. Although cancer cells survive with increased mutagenic capacity, they are also more dependent on mechanisms that ensure genome stability in order to abrogate catastrophic DNA damage [28]. Thus, targeting this hallmark of cancer cells has emerged as a promising therapeutic strategy. Here we show that the transcriptional program driven by E2F is key for the correct temporal control of the cell cycle progression and for the preservation of genome integrity in PCa cells.

The role of E2F activity in maintaining a balance between increased proliferation capacity and sustained DNA damage may underlie the addiction of some tumor cells to abnormally high levels of E2F activity. In fact, increased E2F expression and E2F-dependent transcriptional profiles have been observed in several cancer types including our work on the prostate and studies on pancreatic and head and neck squamous cell carcinoma [36,37]. In prostate cancer, it has been proposed that elevated E2F1 expression might contribute to the progression of hormone-independent PCa through its ability to repress the expression of the androgen receptor [38,39]. Our analyses with PCa cell lines as well as with prostate tumor specimens derived from patients suggest that E2F activity might also contribute to PCa malignancy through the production of adequate levels of nucleotides provided by E2F transcriptional targets of the de novo and salvage pathways of nucleotide biosynthesis (Appendix A).

Nucleotide shortage is thought to be a common source of replication stress. Determining how tumor progression may be inhibited by interfering with nucleotide metabolism has received increasing attention [21,40,41,42], although drug resistance has greatly affected the clinical use of this strategy [24]. Mechanisms of 5-FU sensitivity and resistance have been intensively investigated but many details are still largely unknown [22]. It has been widely accepted that TS expression is the main molecular mechanism governing 5-FU sensitivity, and targeting TS has been proposed as an attractive strategy to reverse 5-FU resistance [43,44]. More recently, a single-cell RNA-seq analysis of colon cancer has uncovered three distinct transcriptome phenotypes upon 5-FU treatment related to apoptosis, cell-cycle checkpoint and stress resistance, which are providing a resource for understanding chemoresistance to 5-FU [45]. In our work, we showed that genetic and chemical E2F inhibition reversed the resistance of PCa cells to 5-FU, in line with the findings of a recent study in various cancer cell lines using a small molecule that inhibits E2F1 [46]. Interestingly, we detected both transient and sustained upregulation of some E2F targets upon 5-FU and demonstrated that the sustained induction of the salvage pathway genes *TK1* and *DCK* upon 5-FU treatment is mediated by a combined E2F1/E2F2 activity. This observation initially led us to ponder a possible mechanism for resistance to 5-FU based on the E2F-dependent upregulation of the enzymes of the salvage pathway of nucleotide biosynthesis. However, while addition of nucleotide precursors fully prevented DNA damage elicited by 5-FU, they did not diminish DNA damage of E2F1/E2F2 knockdown cells, indicating that, apart from nucleotide biosynthesis, additional mechanisms are governed by E2F to restrain DNA damage in S phase.

Premature CDK1 activation has been linked with DNA damage in replicating cells by promoting the breakage of replication forks [47]. Nevertheless, the impact that restricting CDK1 activity during S phase might have in the malignant phenotype of prostate cancer has not been addressed. Here we present evidence that combined silencing of E2F1 and E2F2 results in inappropriate CDK1 activation during S phase, concomitantly with reduced expression of *WEE1*, an E2F target gene encoding the CDK1/2 inhibitory kinase WEE1 [19]. Widespread activation of CDK1 throughout the cell cycle has been shown to lead to DNA damage and is toxic for mammalian cells [48], which underscores the relevance of identifying mechanisms that limit CDK1 activity out of mitosis to understand genome integrity maintenance. We here propose E2F activity as one of such arms that restricts CDK1 activity in S-phase by regulating the expression of WEE1. We showed that inhibition of CDK1 prevents E2Fi-induced DNA damage in PC3 cells. Similarly, inactivation of CDK1 in cells treated with CHK1 inhibitors significantly improves DNA damage and proliferative potential of cells [49], and alleviates the genotoxicity imposed by USP7 inhibitors [48]. Restriction of CDK2 by WEE1 has also been shown to ensure genome maintenance during DNA replication [17,30]. However, inhibition of CDK2 in PC3 cells did not rescue DNA damage of E2F1/2 targeted cells. In DU145 cells, we uncovered the opposite scenario, by which limiting CDK2 activity and not CDK1 is the main mechanism to preserve genome integrity, suggesting different dependencies on CDK1 and CDK2 activities in PC3 and DU145 cell lines. Although we have not tested, the relative abundance of CDK1 and CDK2 in the cells could account for the different dependencies, as previously reported [32].

E2F activity has been linked to the transcriptional control of several mitotic genes [1]. However, E2F inhibition did not increase the mitotic index, and thus, differences in mitosis do not seem to account for the observed increased CDK1 activity upon E2F1/2 loss. Instead, our data suggest that E2F1/2 restrict CDK1/2 activation and control DNA stability through the transcriptional regulation of their target gene *WEE1*. Future research will provide mechanistic data on the implication of WEE1 in the control of genome integrity governed by E2F1/2, as well as the role of CDC25A and CDC25C phosphatases in this novel function of E2F. Interestingly, a recent study revealed that in addition to its role in mitosis, CDK1 is also active during G1/S to allow S-phase entry, demonstrating that CDK1 regulates two distinct transitions of the cell cycle [50]. Whether the E2F-WEE1 axis controls this distinct function of CDK1 deserves further attention.

We show that genome instability of E2F1/E2F2-targeted cells fires ATR-mediated replication stress response. In this scenario, ATR could be targeted to induce catastrophic levels of DNA damage that would redirect the cells to apoptosis [35]. Indeed, E2F and ATR compound inhibition boosts replication stress and dramatically reduces colony forming capacity and xenograft growth of PCa cells. Our data are consistent with reported work showing increased sensitivity to ATR inhibition in the context of replication stress [20,51]. We show that inhibition of CDK1 prevents the increase in DNA damage in PC3 cells co-treated with E2Fi and ATRi, pinpointing the enhanced CDK1 activity as the mechanism responsible for the DNA damage inflicted by the E2Fi-ATRi combinatorial treatment. Although we have not tested, considering that the addition of nucleotide precursors did not diminish DNA damage of E2F-depleted cells, dNMP shortage is not likely responsible for the increased DNA damage after E2Fi-ATRi combinatorial treatment. Instead, enhanced CDK1 activity emerges as a mechanism accounting for DNA damage in this scenario. ATR inhibition has already shown benefits for the treatment of PCa in combination with PARP inhibitors [52]. Based on our data, inhibition of E2F activity emerges as a promising strategy to provoke catastrophic levels of replication stress in combination with drugs targeting ATR, and this combination might be explored in the clinical setting for the treatment of PCa.

## 5. Conclusions

E2F1/E2F2 safeguard genome integrity by promoting nucleotide biosynthesis and restraining CDK1 activity. Inhibition of E2F in combination with drugs targeting nucleotide biosynthesis or DNA repair is a promising strategy that could be applied to PCa treatment.

## Figures and Tables

**Figure 1 cancers-14-04952-f001:**
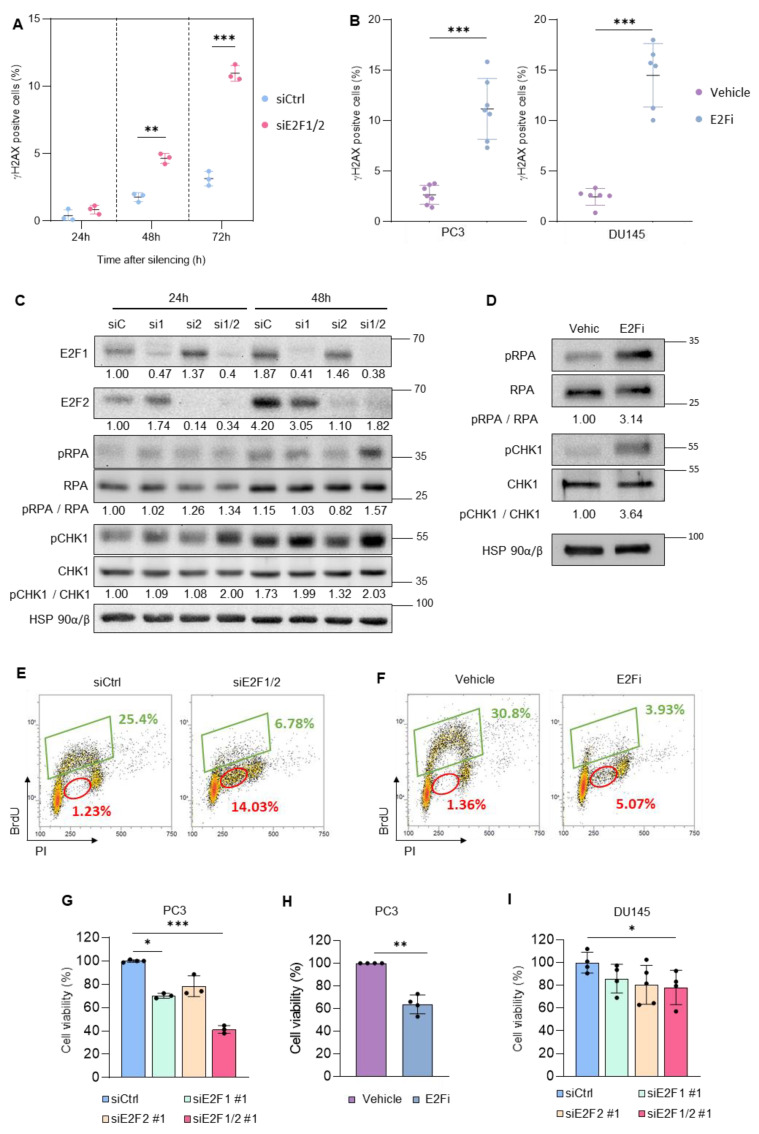
Genetic or chemical inhibition of E2F1/2 induces replication stress and compromises cellular viability. (**A**) Percentage of γH2AX-positive cells at the indicated time points after transfection with siCtrl or siE2F1/2#1. Data shows the average ± SD from 3 independent experiments. (**B**) Percentage of γH2AX-positive cells 24 h after treatment with E2Fi in PC3 (20 µM, left panel) and DU145 cells (40 µM, right panel). In PC3 cells, data show the average ± SD from 7 independent experiments. In DU145 cells, data show the average ± SD from 6 independent experiments. (**C**) Representative Western blot analysis of E2Fs and several replication stress response activation markers in extracts prepared from PC3 cells 24 or 48 h after transfection with siCtrl (sC), siE2F1#1 (si1), siE2F2#1 (si2), or siE2F1/2#1 (si1/2). HSP90 was used as loading control. Densitometric values were used to calculate phospho-to-total protein ratios, expressed as fold-over 24 h siCtrl sample. (**D**) Representative Western blot analysis of RPA and CHK1 in extracts prepared from PC3 cells 8h after treatment with E2Fi (20 µM). HSP90 was used as loading control. Densitometric values were used to calculate phospho-to-total RPA ratios, expressed as fold-over vehicle sample. (**E,F**) Representative cell cycle distribution analysis of PC3 cells 72 h after transfection with siCtrl or siE2F1/2#1 (**E**) or after 24 h treatment with E2Fi (20 µM) (**F**). Cells were pulse labeled with BrdU for the last 2 h, harvested and fixed. DNA content was analyzed by staining with PI and DNA synthesis was assessed by staining with an antibody to BrdU and measured by flow cytometry. Percentages of BrdU-positive cells are shown in green and BrdU-negative cells with 2C–4C DNA content are shown in red. (**G**) Quantification of PC3 cell number using Trypan blue exclusion dye 96 h after transfection with non-target control siRNAs (siCtrl), with siRNAs specific for E2F1 (siE2F1 #1), E2F2 (siE2F2 #1) or for E2F1 and E2F2 (siE2F1/2 #1). Results of cell viability are expressed as percentage of viable cells over the siCtrl vehicle sample. (**H**) Quantification of PC3 cell number using Trypan blue 24 h after the treatment with E2Fi (20 µM) or vehicle. (**I**) Quantification of DU145 cell number using Trypan blue 72 h after transfection with non-target control siRNAs (siCtrl), with siRNAs specific for E2F1 (siE2F1 #1), E2F2 (siE2F2 #1) or for E2F1 and E2F2 (siE2F1/2 #1). Results of cell viability are expressed as percentage of viable cells over the siCtrl sample. Statistical analysis was performed using ANOVA and Fisher’s test. *** *p* < 0.0001, ** *p* < 0.001, * *p* < 0.05.

**Figure 2 cancers-14-04952-f002:**
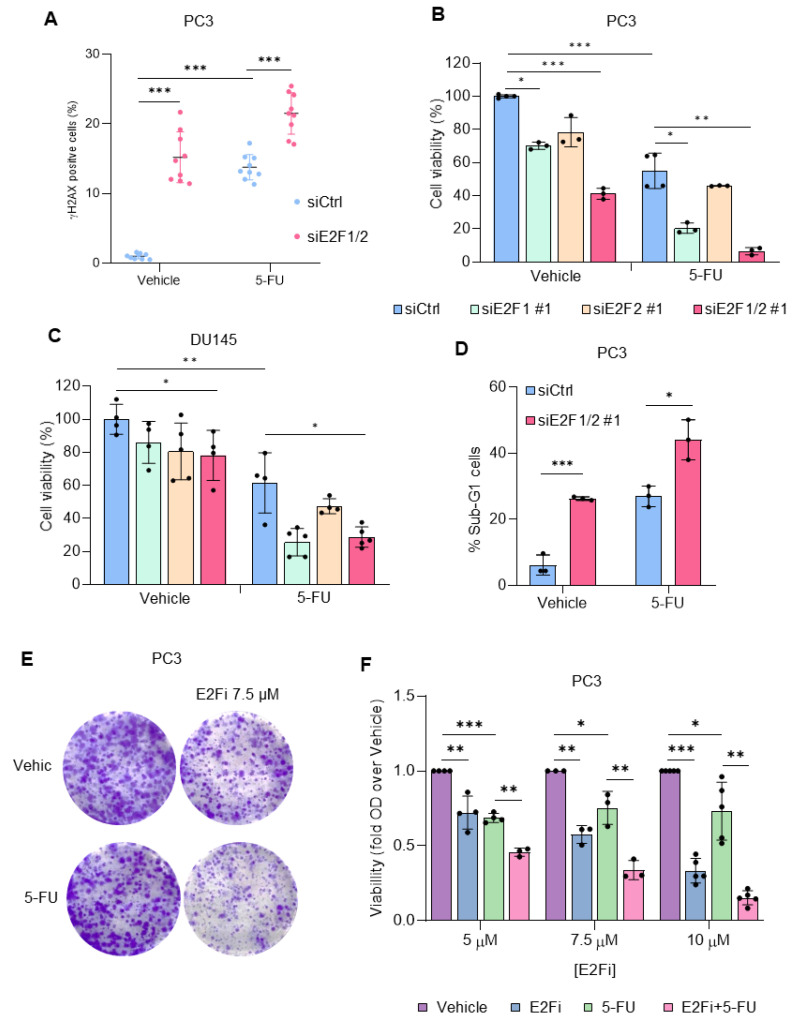
E2F knockdown or chemical inhibition potentiates 5-FU-elicited effects. (**A**) Percentage of γH2AX-positive cells 72 h after transfection with siCtrl or siE2F1/2#1 treated with vehicle or 5-FU during the last 24 h. Data from 9 independent experiments are shown. (**B**) Quantification of PC3 cell number using Trypan blue exclusion dye 96 h after transfection with non-target control siRNAs (siCtrl), with siRNAs specific for E2F1 (siE2F1 #1), E2F2 (siE2F2 #1) or for E2F1 and E2F2 (siE2F1/2 #1) and treatment with 5-FU (5 µM) for the last 72 h. Results of cell viability are expressed as percentage of viable cells over the siCtrl vehicle sample. (**C**) Quantification of DU145 cell number using Trypan blue 72 h after transfection with non-target control siRNAs (siCtrl), with siRNAs specific for E2F1 (siE2F1 #1), E2F2 (siE2F2 #1) or for E2F1 and E2F2 (siE2F1/2 #1) and treatment with 5-FU (5 µM) for the last 48 h. Results of cell viability are expressed as percentage of viable cells over the siCtrl vehicle sample. (**D**) Percentage of total cells with DNA content compatible with apoptosis 96 h after transfection with non-target control siRNAs (siCtrl) or with siRNAs specific for E2F1 and E2F2 (siE2F1/2 #1) and treatment with 5-FU (5 µM) for the last 72 h. Data show the average ± SD from 3 independent experiments. (**E**) Colony forming capacity of PC3 cells after treatment with 5-FU and E2Fi. PC3 cells were treated with 5-FU (1 µM) and E2Fi (5, 7.5 and 10 µM) alone or in combination. 72 h later, drugs were washed out and cells were incubated with fresh media for an additional 10–12 days. Then, cells were fixed and stained with crystal violet. Representative images of colony density obtained with 5-FU and 7.5 µM E2Fi are shown. (**F**) Quantification of CFA represented in (**E**). Crystal violet was dissolved with acetic acid and the absorbance was measured by spectrophotometry. Relative cell viability was calculated by normalizing the absorbance of each condition using their vehicle controls. Statistical analysis was performed using ANOVA and Fisher’s test. *** *p* < 0.0001, ** *p* < 0.001, * *p* < 0.05.

**Figure 3 cancers-14-04952-f003:**
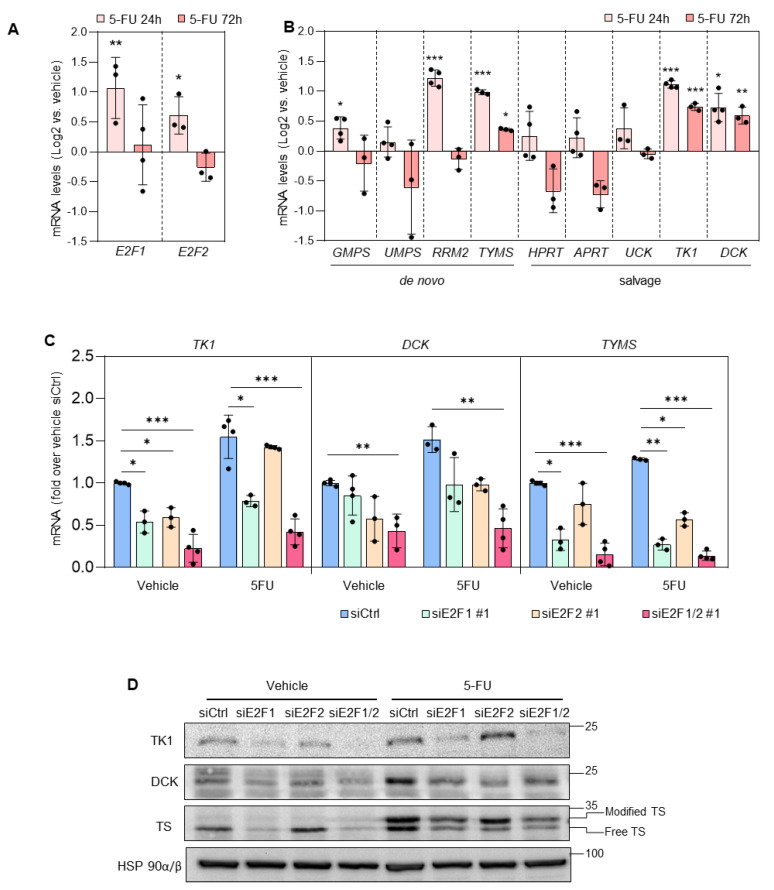
E2F1 and E2F2 are necessary for 5-FU-mediated sustained upregulation of *TYMS*, *TK1* and *DCK*. (**A**,**B**) RT-QPCR was carried out to analyze the expression of E2F1 and E2F2 (**A**) and their target genes involved in nucleotide biosynthesis (**B**) in PC3 cells after 24 and 72 h of 5-FU (5 µM) treatment compared with sample treated with vehicle. (**C**) RT-QPCR analysis of *TK1*, *DCK* and *TYMS* in PC3 cells 96 h after transfection with non-target control siRNAs (siCtrl), siRNAs specific for E2F1 (siE2F1#1), E2F2 (siE2F2#1) or with their combination (siE2F1/2#1) and treated for the last 72 h with vehicle or 5-FU (5 µM). In (**A**–**C**), *L19* mRNA was used as the normalization control. The values are expressed as Log2 ratio vs. vehicle control (**A**,**B**) or fold over siCtrl (**C**) and are the results (mean ± SD) of three independent experiments. Statistical analysis was performed using ANOVA and Fisher’s test. *** *p* < 0.0001; ** *p* < 0.001, * *p* < 0.05. (**D**) Representative Western blot analysis of TK1, DCK and TS in extracts prepared from PC3 cells 96 h after transfection with siCtrl, siE2F1#1, siE2F2#1, or siE2F1/2#1 and treated for the last 72 h with 5-FU (5 µM). HSP90 was used as loading control. Note the presence of 5-FU-modified TS (inactive) in treated samples.

**Figure 4 cancers-14-04952-f004:**
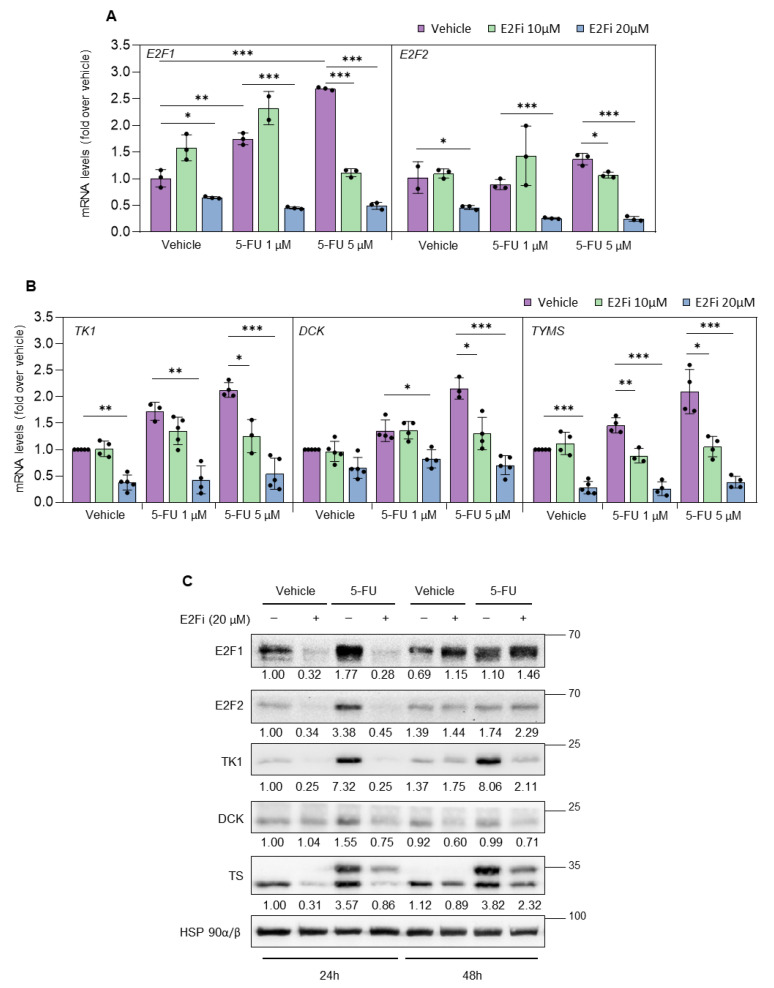
E2F inhibitor HLM (E2Fi) downregulates basal levels of E2F1, E2F2, TK1, DCK and TS (*TYMS*) and prevents 5-FU-induced upregulation. RT-QPCR analysis of *E2F1* and *E2F2* (**A**) and *TK1*, *DCK* and *TYMS* (**B**) in PC3 cells 24 h after co-treatment with E2Fi (10, 20 µM) and 5-FU (1, 5 µM). *L19* mRNA was used as normalization control. Results are expressed as fold-over samples treated with vehicle (mean ± SD) from 3 to 4 independent experiments. Statistical analysis was performed using ANOVA and Fisher’s test. *** *p* < 0.0001, ** *p* < 0.001, * *p* < 0.05. (**C**) Representative Western blot analysis of E2F1, E2F2, TK1, DCK and TS in extracts prepared from PC3 cells 24 or 48 h after co-treatment with E2Fi (20 µM) and 5-FU (5 µM). Expression of HSP90 was used as loading control. Numbers below the bands correspond to the relative densitometric values, expressed as fold-over vehicle 24 h sample. Similar results were obtained in at least 4 independent experiments.

**Figure 5 cancers-14-04952-f005:**
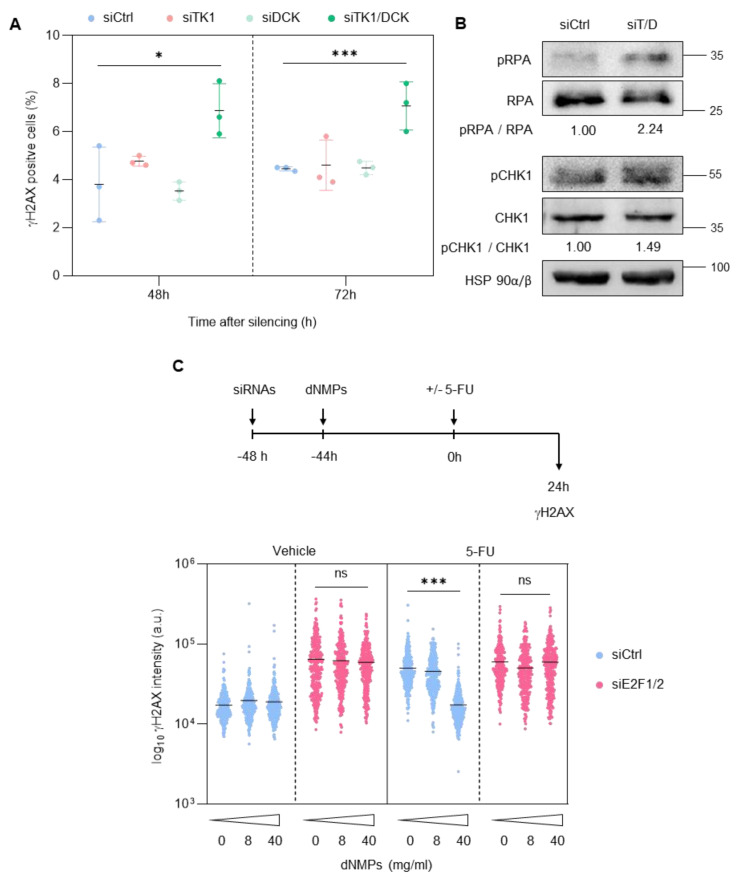
Combined TK1 and DCK activity contributes to maintain genomic stability but fails to rescue DNA damage of E2F1/E2F2 knockdown cells. (**A**) Percentage of γH2AX-positive cells at the indicated time points after transfection with siCtrl, siTK1, siDCK or siTK1/DCK. Data shows the average ± SD from 3 independent experiments. (**B**) Representative Western blot analysis of several replication stress response markers in extracts prepared from PC3 cells 72 h after transfection with siCtrl or siTK1/DCK (siT/D). HSP90 was used as loading control. Densitometric values were used to calculate phospho-to-total protein ratios, expressed as fold-over siCtrl sample. (**C**) Schematic representation of the experimental design. PC3 cells were co-transfected with siE2F1 and siE2F2 with set #1 oligos. Then, 4 h later, cells were treated with 0, 8, 40 mg/mL of dNMP, and 20 h later, treated with 5-FU (5 µM). Intensity of γH2AX signal in cells in S phase 24 h after the indicated treatments is represented. Data are expressed as Log10 intensity. Horizontal bars mark mean intensity. Data of 300 cells from one representative experiment of three independent experiments are shown. Statistical analysis was performed using ANOVA and Fisher’s test. *** *p* < 0.0001, * *p* < 0.05, ns = non-significant.

**Figure 6 cancers-14-04952-f006:**
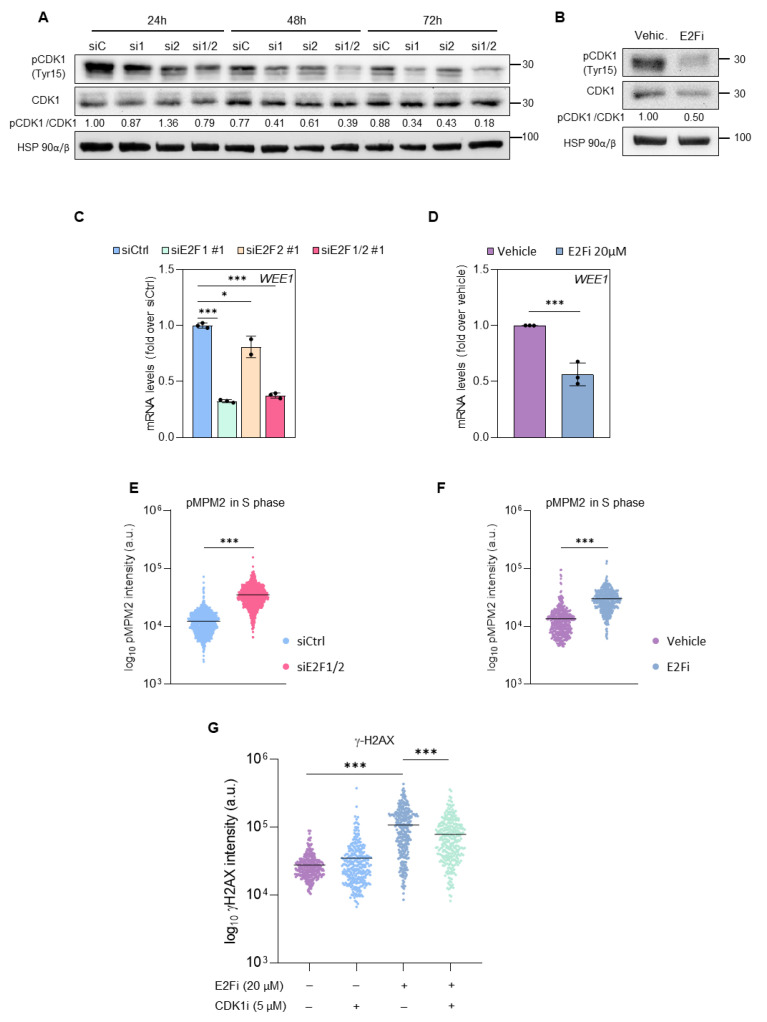
E2F1/2 knockdown or E2Fi increases CDK1 activity, and inhibition of CDK1 with RO3306 (CDK1i) prevents E2Fi-induced DNA damage in PC3 cells. (**A**,**B**) Representative Western blot analysis of CDK1 phosphorylated at Tyr 15 (pCDK1) and total CDK1 in extracts prepared from PC3 cells 24, 48 or 72 h after transfection with siCtrl, siE2F1#1, siE2F2#1, or siE2F1/2#1 (**A**) or treated with E2Fi (20 µM) for 24 h (**B**). HSP90 was used as loading control. Densitometric values were used to calculate relative levels of pCDK1/CDK1, expressed as fold-over vehicle siCtrl (**A**) or vehicle (**B**) samples. (**C**,**D**) RT-QPCR analysis of *WEE1* in PC3 cells 24 h after transfection with siCtrl, siE2F1#1, siE2F2#1 or siE2F1/2#1 (**C**) or 24 h after treatment with E2Fi 20 µM (**D**). *L19* mRNA was used as normalization control. Results are expressed as fold-over 24 h siCtrl, or vehicle (mean ± SD) from 1–3 independent experiments done in triplicate. (**E**,**F**) Intensity of pMPM2 signal in cells in S phase 72 h after transfection with siCtrl or siE2F1/2#1 (**E**), or 24 h after treatment with E2Fi (20 µM, (**F**)). Horizontal bars mark mean intensity. Data of 1000 cells from one representative experiment of 3 independent experiments are shown. (**G**) Intensity of γH2AX signal in cells in S phase 24 h after the indicated treatments. Data are expressed as Log10 intensity. Horizontal bars mark median intensity. Data of 300 cells from one representative experiment of three independent experiments are shown. Statistical analysis was performed using ANOVA and Fisher’s test. *** *p* < 0.0001, * *p* < 0.05.

**Figure 7 cancers-14-04952-f007:**
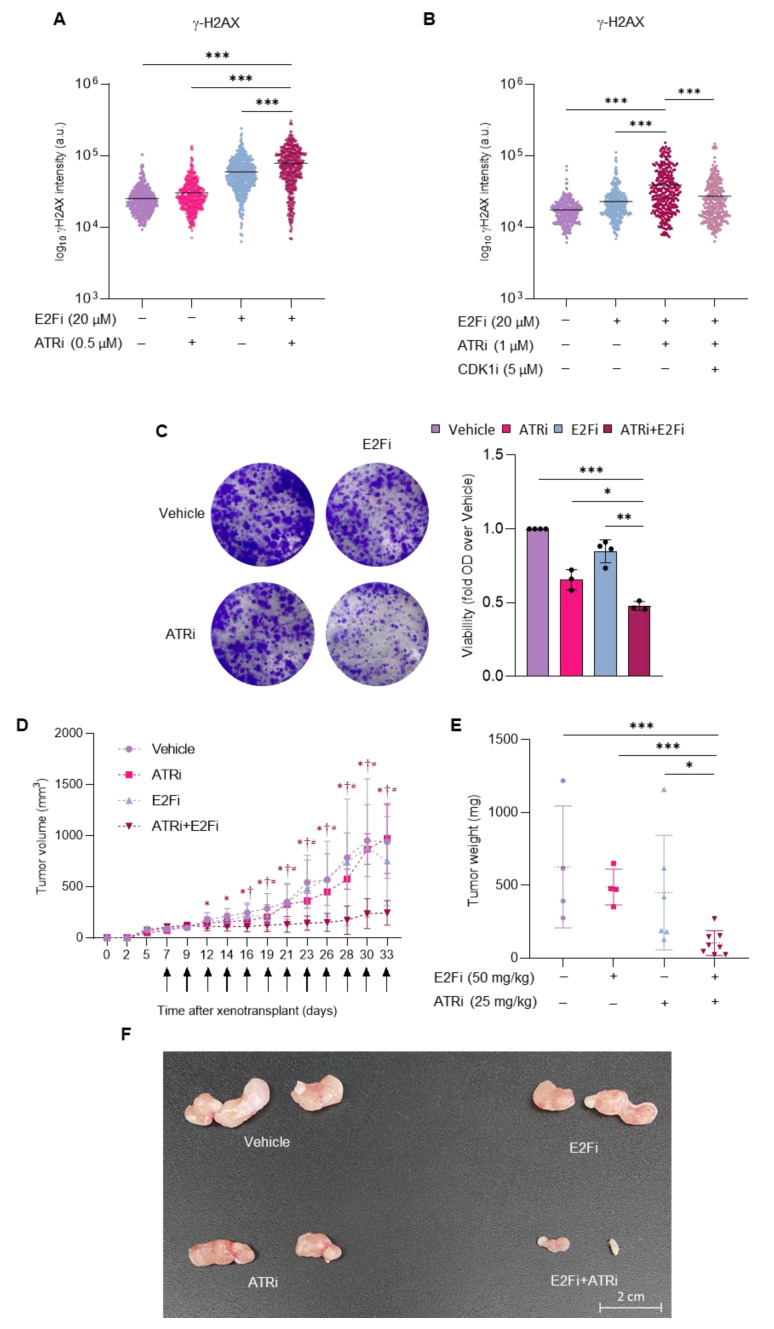
Combination of E2Fi and ATRi enhances genomic instability and reduces cell growth in PC3 cells in vitro and in xenografts. (**A**) Intensity of γH2AX signal in cells in S phase 24 h after the indicated treatments. Data of 500 cells from one representative experiment of 3 independent experiments are shown. Data are expressed as Log10 intensity. Horizontal bars mark mean intensity. Statistical analysis was performed using ANOVA and Fisher’s test. *** *p* < 0.0001. (**B**) Intensity of γH2AX signal in cells in S phase 24 h after the indicated treatments. Data of 300 cells from one representative experiment of 3 independent experiments are shown. Data are expressed as Log10 intensity. Horizontal bars mark mean intensity. Statistical analysis was performed using ANOVA and Fisher’s test. *** *p* < 0.0001. (**C**) Representative images of colony density in each condition. PC3 cells were treated with E2Fi (5 µM) and ATRi (0.1 µM) alone or in combination. 24 h later, drugs were washed out and cells were incubated with fresh media for an additional 10–12-day period. Then, cells were fixed and stained with crystal violet. Graph shows the quantification of CFA. Crystal violet was dissolved with acetic acid and the absorbance was measured by spectrophotometry. Relative cell viability was calculated by normalizing the absorbance of each condition using their vehicle controls. Statistical analysis was performed using ANOVA and Fisher’s test. *** *p* < 0.0001, ** *p* < 0.001, * *p* < 0.05. (**D**) PC3 xenograft growth after treatment with E2Fi and ATRi. PC3 cells were injected subcutaneously into both flanks in 20 CD-1 nude mice. When tumor volume reached 65 mm^3^, mice were divided in 4 groups (Vehicle, E2Fi, ATRi and combination). E2Fi and ATRi were administered IP at 50 mg/kg and 25 mg/kg, respectively. Growth curves depict mean (±SD) of the tumor volume. Arrows indicate the days of injection. n = 4–8 tumors per group. Statistical analysis was performed using ANOVA and Fisher’s test. * *p* < 0.05 vs. vehicle, † *p* < 0.05 vs. E2Fi, # *p* < 0.05 vs. ATRi. (**E**) Weight of the tumors obtained at the endpoint from mice treated as indicated. n = 4–8. Statistical analysis was performed using ANOVA and Fisher’s test. *** *p* < 0.0001, * *p* < 0.05. (**F**) Representative tumors from control and experimental mice are presented.

## Data Availability

The datasets supporting the conclusions of this article are included within the article (and Appendix A).

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
