# Peer review of "Targeting E2F Sensitizes Prostate Cancer Cells to Drug-Induced Replication Stress by Promoting Unscheduled CDK1 Activity"

_cancers, 2022, doi:10.3390/cancers14194952_

Round 1

Reviewer 1 Report (Previous Reviewer 1)

I am satisfied with the corrections made with the paper and would support its publication.

Reviewer 2 Report (Previous Reviewer 3)

The authors have successfully answered all my questions. The revised version of the manuscript is well written, the conclusions are clear, and I recommend the acceptance of the manuscript for publication.

This manuscript is a resubmission of an earlier submission. The following is a list of the peer review reports and author responses from that submission.

Round 1

Reviewer 1 Report

In this paper the authors show that chemical and genetic inhibition of E2F1 and E2F2 shows an increase in replication stress through H2AX phosphorylation. The evidence supporting this is well documented and supports the main claims of the paper. The downstream analysis of transcription of E2F target genes involved in nucleoside biogenesis leads to identification of the potential role of CDK1 activity in promotion of DNA replication stress and the synergistic effects of ATRi and E2Fi in reducing tumour volume. I found the paper to be scientifically rigorous but there were some details missing as described below.

1)     The statistical tests are described in the methods, but this information should be included in legends of each figure where statistical analysis is performed.

2)     The figures are presented in blocks (figures 3-5 and Figures 6 and 7) such that the description of results is typically far away from the source data. Try to place the figures closer to the relevant text.

3)     In Figure 6, there are 7 panels and 4 labels (A-D) reformat showing A-G as I found this confusing in current format.

4)     In Figure 6C (as currently labelled) there is evidence of a CDK1 specific phosphorylation of MPM2. It was stated in text that this is S-phase specific labelling. I assume that this from flow cytometry as there are no cell cycle synchronisation approaches. Please put in the gated data figure for the selection of the S-phase population in supplementary figures. In addition, western blotting is much preferred for quantitation of phosphospecific antibodies. Please show more evidence that there is a differential CDK1 activity under these conditions as this result is central to the conclusions of this work as presented.

5)     Minor point in line 215 there is a type, remove “de” and replace with “the”

Reviewer 2 Report

In the manuscript entitled with “targeting E2F sensitizes prostate cancer cells to drug-induced replication stress buy promoting unscheduled CDK1 activity”, the authors give a good attempt to study the contribution of E2F, which is highly expressed in many cancer cell types, to malignancy. The manuscript is well written and the experiments design is rational and the data were well presented.

I have the minor comments as following:

1.     In the figure 1 C, the authors checked the level of both RPA &pRPA and Chk1&pChk1 in cells treated with siRNAs targeting E2F, I wonder why in the figure 1C, when cells were treated with E2F inhibitor, only  RPA &pRPA was checked? How about Chk1&pChk1?

2.     In figure 2B and 2C, it will be easier to label the cell line name “PC3” and “DU145” respectively in the figure to make it easier to read. 

Reviewer 3 Report

In this study, Hamidi M et al., show that E2F1/E2F2 inhibition induces DNA damage in Prostate Cancer (PCa) cell lines and facilitates therapeutic response to drugs that induce replication stress. E2F inhibition reduces the expression of E2F-target genes involved in nucleotide biosynthesis and mediates CDK1/CDK2 premature activation by reducing the levels of WEE1. dNMPs do not rescue the E2F inhibition-induced DNA damage, while the CDK1 or CDK2 inhibitors humbly decrease DNA damage after E2F inhibition. The E2F inhibition-induced DNA damage synergizes with the ATR inhibitor to reduce cell growth and tumor volume in PCa cell lines and xenograft models.

The authors executed well-designed experiments and presented clear data to provide mechanistic insight into how E2F activity protects PCa cells from DNA damage when they are treated with cytotoxic drugs. More importantly, the authors suggest a new combination treatment for E2F amplified PCa that targets DNA repair mechanisms. Although some data require further analysis to provide a more unambiguous interpretation of the E2F-mediated mechanism in DNA repair, I recommend this manuscript for publication in Cancers. There is one major point that, if the authors can answer, will improve the impact of the study:

Major point:

1.     Do the synergistic (or additive) activities between E2Fi and 5-FU and between E2Fi and ATRi depend on dNMPs or CDK1/CDK2?

Minor points:

1.     Discuss the difference between the different time points in the E2F-mediated transcription regulation of the genes involved in nucleotide biosynthesis (Fig. 3). Can you provide the data for both 24h and 72h for the experiments presented in Fig. 3? Label the graphs to indicate the timepoint when the samples were collected.

2.     Data in Fig.5C, 6D, and 7A should be presented similarly (intensity or % of positive cells). Otherwise, it is difficult for the reader to make direct comparisons and have clear conclusions.

3.     In Fig. 7A, 7B, and 7D, please indicate the p-value between the 2nd and 3rd and between the 3rd and 4th samples. This will help the reader to understand the difference between single treatment and combination.

4.     Label the graphs in Fig.2 to make clear what graph corresponds to which cell line.

5.     Lane 315: “panel F” needs correction.